# Robust Self-reflective Hashing for Cross-modal Retrieval with Noisy Label

**Hao Sun** [1]   **Qibing Qin** [2]   **Lei Huang** [3]

## Abstract

Cross-modal Hashing (CMH) typically assumes a perfectly complete data annotation, whereas noisy labels are unavoidable in practical scenarios. Existing CMH methods often overlook the uncertainty introduced by noise or semantic ambiguity, making models susceptible to overfitting noisy labels and yielding unreliable similarity judgments during inference. To address this issue, we propose a Robust Self-reflective Hashing (RSH) framework that prudently analyzes semantic discrepancies while accounting for uncertainty, thereby effectively mitigating interference from noisy labels. Specifically, the Double Feature Representation (DFR) method is introduced, employing semantic and uncertainty features to represent the semantic representation and fuzziness of samples. With a double feature, we propose a novel cross-modal similarity metric - the Self-reflective Similarity Metric (SSM), which judges sample similarity by integrating semantic discrepancy and fuzziness, enabling the model to adaptively weaken semantic discrepancy according to uncertainty level. The proposed method is plug-and-play, enabling seamless integration into diverse objective functions to enhance model robustness and reliability. Extensive experiments on benchmark datasets demonstrate that RSH outperforms existing methods. Code is available at `https://github.com/QinLab-WFU/RSH`.

## 1. Introduction

Cross-modal hashing (CMH) has emerged as an effective solution for large-scale multi-modal retrieval tasks owing to its

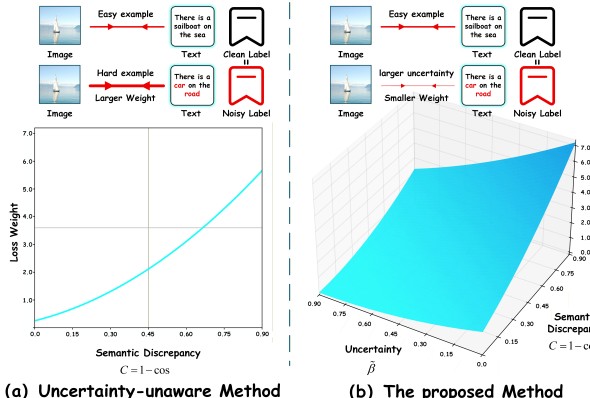

*Figure 1.* Illustration of the motivation behind the proposed robust self-reflective hashing. (a) Uncertainty-unaware methods impose constraints on samples solely based on semantic discrepancy, while ignoring the uncertainty induced by noise or semantic ambiguity. (b) Our method explicitly captures the uncertainty of heterogeneous samples and adaptively weakens semantic discrepancy according to the uncertainty level, thereby alleviating the adverse influence of noisy labels on the learning of hash codes.

low storage costs and high retrieval efficiency (Chen et al., 2022; Sun et al., 2023; Wang et al., 2025; Pu et al., 2025a). Under ideal conditions and supported by high-quality multimodal training data, CMH methods can achieve outstanding retrieval performance (Cheng et al., 2026; Huo et al., 2024b; Qin et al., 2025; Sun et al., 2024). However, existing approaches largely implicitly assume that all training samples are completely labeled, which is difficult to achieve in practice — manual annotation or annotation by non-specialists inevitably generates noisy labels (Pu et al., 2025b; Wang et al., 2024; Hu et al., 2021). These noisy labels misguide the model's similarity learning process, leading to a significant decline in retrieval performance. Consequently, enhancing model robustness and reliability under noisy label conditions has become an urgent challenge.

To mitigate the impact of noisy labels, existing research has proposed various strategies for CMH tasks, including label refinement (Wang et al., 2026), robust loss designs (Hu et al., 2021; Pu et al., 2025b), and early learning regularization (Xu et al., 2022). Although these approaches mitigate the adverse effects of noisy labels to some extent, they solely employ a semantic feature to represent samples, thereby overlooking the inherent uncertainty introduced by the noise

[1]School of Computer Science, Qufu Normal University, Rizhao, China [2]School of Computer Engineering, Weifang University, Weifang, China [3]Faculty of Information Science and Engineering, Ocean University of China, Qingdao, China. Correspondence to: Qibing Qin <qinbing@wfu.edu.cn>.

*Proceedings of the 43$^{rd}$ International Conference on Machine Learning, Seoul, South Korea. PMLR 306, 2026. Copyright 2026 by the author(s).*

or semantic ambiguity. This renders models ill-equipped to adapt to the dynamic shifts in sample uncertainty induced by noise or semantic ambiguity, as illustrated in Figure 1 (a).

To perceive sample uncertainty, researchers have proposed various probabilistic embedding methods that capture sample uncertainty through distribution modeling (Cheng et al., 2025; Han et al., 2026). These approaches typically use KL divergence (Hershey & Olsen, 2007) or Monte Carlo-based distance (Oh et al., 2018) to measure sample similarity, treating variance as a measure of uncertainty. However, they still perform similarity calculations and gradient updates using uncertainty-assisted samples, which means they fail to effectively suppress misleading training signals generated by noisy labels (Hu et al., 2025; Wang et al., 2023).

This paper argues that models should focus more on certain samples and actively reduce the influence of uncertain samples. This mirrors human learning mechanisms — individuals typically disregard unreliable information to avoid disrupting the learning process (e.g., medical experts would never adopt highly uncertain diagnostic results). Consequently, for samples affected by noisy labels, a reasonable strategy is to soften semantic discrepancies based on their uncertainty levels and treat them as possessing a degree of similarity with other samples, as illustrated in Figure 1 (b).

To this end, we propose the Robust Self-reflective Hashing (RSH) framework, which prudently analyzes semantic discrepancies while accounting for uncertainty, thereby effectively suppressing interference from noisy labels or semantic ambiguity. Figure 2 illustrates the overall architecture of the RSH framework. Specifically, we devise a double feature representation that simultaneously generates semantic and uncertainty feature vectors for each sample, capturing its semantic information and uncertainty, respectively. Building upon this, we introduce a self-reflective similarity metric that directly incorporates uncertainty into the similarity calculation process, adaptively softening semantic discrepancies according to the sample's uncertainty level. Furthermore, gradient analysis of the loss function shows that RSH enables the model to learn at a rate that is sensitive to uncertainty. In summary, the main contributions of this paper are as follows:

- We introduce a double feature representation where each sample is jointly represented by a semantic and uncertainty feature, capturing its semantic information and uncertainty level, respectively.

- We propose a novel cross-modal similarity metric — the self-reflective similarity metric, which directly incorporates uncertainty into the similarity calculation procedure and adaptively weakens semantic discrepancy based on uncertainty levels.

- RSH is highly versatile and can be seamlessly integrated into diverse objective functions, enabling models to learn at a pace that is adaptive to uncertainty.

- Extensive experiments validate the robustness and reliability of the proposed RSH in noisy label scenarios, particularly at high noise rates.

## 2. Related Work

### 2.1. Cross-modal Retrieval with Noisy Label

Noisy labels are prevalent in multi-modal datasets due to incomplete annotations or automated labeling. To mitigate the negative impact of noisy labels, existing research has proposed various robust learning strategies, typically categorized into three approaches: label refinement, robust loss designs, and early learning regularization. For instance, Wang *et al.* proposed Noise-Robust Generative Hashing (NRGH), which utilizes Gaussian mixture modeling to identify unreliable samples and dynamically refine noisy labels during optimization (Wang et al., 2026). Wang *et al.* introduced Noise Resistance Cross-modal Hashing (NRCH), which employs a robust contrastive hashing loss to constrain homologous sample pairs rather than noisy positive samples (Wang et al., 2024). Pu *et al.* proposed Robust Self-paced Hashing (RSHNL), which uses a noise-tolerant self-paced loss function to dynamically estimate sample difficulty and progressively distinguish clean from noisy labels (Pu et al., 2025b). Xu *et al.* proposed Early Learning Regularized Contrastive Learning (ELRCMR), which employs early learning regularization to prevent the memorization of noisy labels and dynamic weight balancing to mitigate clustering drift (Xu et al., 2022).

Although these approaches demonstrate robustness in noisy label scenarios, they fail to capture the sample-level uncertainty introduced by noisy labels or semantic ambiguity, leaving models susceptible to misleading training signals. There is still little research on how to mitigate the impact of noise during training by modeling sample uncertainty when dealing with multi-modal data containing noisy labels.

### 2.2. Uncertainty Learning in Cross-modal Retrieval

Recently, uncertainty modeling has been introduced into cross-modal retrieval tasks to enhance model reliability (Gawlikowski et al., 2023). For instance, Han *et al.* proposed Deep Uncertainty-aware Probabilistic Hashing (DU-aPH), which explicitly models sample uncertainty by mapping heterogeneous samples onto multivariate Gaussian distributions (Han et al., 2026). Cheng *et al.* proposed Deep Probabilistic Binary Embedding (DPBE), which employs a Bayesian encoder based on the Laplace approximation to model network weights, thereby capturing uncertainty during the representation learning process (Cheng et al., 2025).

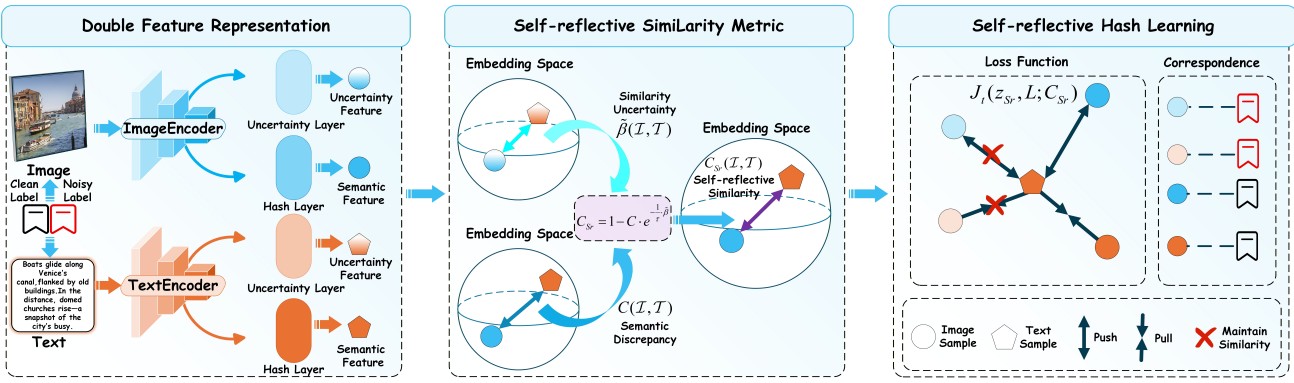

*Figure 2.* Overview of the Robust Self-reflective Hashing (RSH) framework. The framework consists of three key components: (1) double feature representation, extracts semantic and uncertainty features for samples. (2) self-reflective similarity metric, which adaptively weakens semantic discrepancies based on uncertainty levels, enabling more cautious similarity judgements. (3) self-reflective hash learning, which is combined with an objective loss function to guide the model in learning robust and reliable hash codes.

In contrast, Li *et al.* proposed Deep Evidence Cross-Modal Hashing (DECH), which employs evidence theory and beta distributions to model uncertainty, directly estimating uncertainty from network outputs (Li et al., 2025).

Although these methods can capture uncertainty, they still perform similarity calculations and gradient updates using uncertain samples, which makes it difficult to effectively suppress noisy labels at the optimization level. RSH adaptively adjusts the similarity between heterogeneous samples based on uncertainty levels, thereby weakening the semantic discrepancies of samples affected by noisy labels.

## 3. Methodology

### 3.1. Notation

Given the training dataset $O = \{o_i, l_i\}_{i=1}^n$, which contains $n$ paired samples $o_i = \{\mathcal{I}_i, \mathcal{T}_i\}$, where $\mathcal{I}_i$ and $\mathcal{T}_i$ denote the image and text sample for the $i$-th instance, respectively, and $l_i$ is the annotated label for $o_i$. The image and text features are represented as $F^{\mathcal{I}} = \{f_i^{\mathcal{I}}\}_{i=1}^n$ and $F^{\mathcal{T}} = \{f_i^{\mathcal{T}}\}_{i=1}^n$, respectively. Hash functions $H^{\mathcal{I}}$ and $H^{\mathcal{T}}$ project heterogeneous features into the discrete embedding space. Uncertainty functions $U^{\mathcal{I}}$ and $U^{\mathcal{T}}$, which are FC layers, map heterogeneous features onto uncertainty features. Since binary optimization is typically an NP-hard problem, a continuous relaxation strategy $tanh(*)$ is introduced to achieve approximate optimization. The function $sign(*)$ converts binary-like codes to binary codes, defined as:

$$\text{sign}(x) = \begin{cases} +1, & x \geq 0 \\ -1, & x < 0 \end{cases} \quad (1)$$

### 3.2. Double Feature Representation

To achieve a similarity metric that is aware of uncertainty, we first need to model the semantic and uncertainty features

of heterogeneous samples jointly. Specifically, building on prior work (Radford et al., 2021), we use a visual transformer and a text transformer as encoders to extract features. Each encoder comprises 12 such blocks, each consisting of a Multi-Header Self-Attention (MHSA) module and a Multi-Layer Perceptron (MLP) module. The feature extraction process for the original input can be represented as follows:

$$f^* = E^*(\theta_*, *), * = \{\mathcal{I}, \mathcal{T}\} \quad (2)$$

where $E$ denotes the encoder and $\theta$ represents the model parameters. We define semantic feature as $h^* = tanh(\mathcal{H}^*(f^*))$. Furthermore, the uncertainty feature is defined as $u^* = U^*(f^*)$. Building on this, this paper jointly employs semantic and uncertainty features to represent samples, as indicated below:

$$z_{Sr} = (h^*, u^*), * = \{\mathcal{I}, \mathcal{T}\} \quad (3)$$

where $h$ and $u$ describe the semantic characteristics and fuzzy degrees of the sample, respectively.

### 3.3. Self-reflective Similarity Metric

Existing CMH methods primarily encode semantic information within the embedding space, ignoring sample uncertainty caused by noise or semantic ambiguity. This can result in models overfitting noisy labels. To address this issue, we propose a Self-reflective Similarity Metric (SSM), which reduces semantic discrepancies based on uncertainty level, thereby preventing unreliable samples from disrupting the model training. To comparing multi-modal samples $\mathcal{I}$ and $\mathcal{T}$, we define the semantic discrepancy as follows:

$$C(\mathcal{I}, \mathcal{T}) = 1 - \cos(h^{\mathcal{I}}, h^{\mathcal{T}}) = 1 - \frac{(h^{\mathcal{I}})^\top h^{\mathcal{T}}}{\|h^{\mathcal{I}}\|_2 \|h^{\mathcal{T}}\|_2} \quad (4)$$

where $\cos(h^{\mathcal{I}}, h^{\mathcal{T}})$ indicates cosine similarity between sample pairs. Additionally, similarity uncertainty is defined

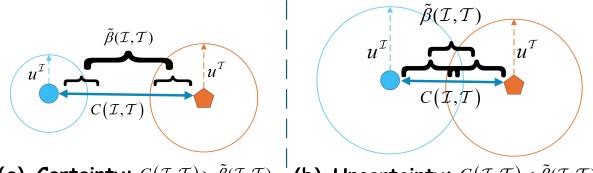

**(a) Certainty:** $C(\mathcal{I},\mathcal{T}) > \tilde{\beta}(\mathcal{I},\mathcal{T})$    **(b) Uncertainty:** $C(\mathcal{I},\mathcal{T}) < \tilde{\beta}(\mathcal{I},\mathcal{T})$

*Figure 3.* Illustration of the RSH under different uncertainty levels. (a) $C(\mathcal{I},\mathcal{T}) > \tilde{\beta}(\mathcal{I},\mathcal{T})$, the semantic relation is considered reliable and contributes effectively to similarity learning. (b) $C(\mathcal{I},\mathcal{T}) < \tilde{\beta}(\mathcal{I},\mathcal{T})$, the influence of semantic discrepancy is adaptively weakened to suppress the effect of noisy labels.

as follows:

$$\beta(\mathcal{I},\mathcal{T}) = \left\| u^{\mathcal{I}} + u^{\mathcal{T}} \right\|_2 \tag{5}$$

Notably, rather than measuring uncertainty for each modality separately, this paper first sums the uncertainty feature vectors before computing the norm. As illustrated in Figure 3, the semantic relationship between heterogeneous samples is deemed unreliable when the following conditions are met:

$$\beta(\mathcal{I},\mathcal{T}) + \gamma \geq C\left(\mathcal{I},\mathcal{T}\right) \tag{6}$$

where $\gamma \geq 0$ represents a self-reflective bias term reflecting the control model's sensitivity to uncertainty. When $\gamma > 0$, the model is cautious in its judgements even when it does not output high uncertainty. Based on the above definition, this paper preliminarily defines SSM as follows:

$$\widetilde{C}_{Sr}(\mathcal{I},\mathcal{T}) = 1 - C(\mathcal{I},\mathcal{T}) \cdot I(C(\mathcal{I},\mathcal{T}) - \beta(\mathcal{I},\mathcal{T}) - \gamma) \tag{7}$$

where $I(x)$ is the indicator function, outputting 1 when $x > 0$ and 0 otherwise. However, the indicator function exhibits discontinuity during optimization, which hinders gradient-based end-to-end training. To address this issue, the concept of relative uncertainty is introduced, which is defined as follows:

$$\widetilde{\beta}(\mathcal{I},\mathcal{T}) = \frac{\beta(\mathcal{I},\mathcal{T}) + \gamma}{\alpha(\mathcal{I},\mathcal{T})} \tag{8}$$

where $\alpha(\mathcal{I},\mathcal{T}) = \left\| h^I - h^T \right\|_2$ regulates the intensity of uncertainty effects. Clearly, relative uncertainty remains non-negative. By weakening semantic discrepancy through relative uncertainty, we define SSM as follows:

$$C_{Sr}(\mathcal{I},\mathcal{T}) = 1 - C(\mathcal{I},\mathcal{T}) \cdot e^{\left(-\frac{1}{\tau} \cdot \widetilde{\beta}(\mathcal{I},\mathcal{T})\right)} \tag{9}$$

where $\tau > 0$ is a hyperparameter that controls the softening degree of semantic discrepancy.

Intuitively, SSM adjusts the similarity between sample pairs by jointly modeling semantic discrepancy and similarity uncertainty. Unlike traditional similarity calculations, which

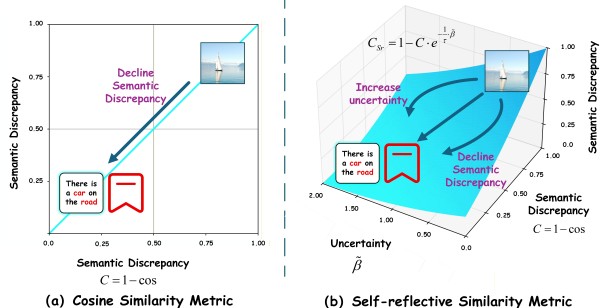

**(a) Cosine Similarity Metric**    **(b) Self-reflective Similarity Metric**

*Figure 4.* Illustration of the proposed SSM compared with the cosine similarity metric. (a) The cosine similarity metric reduces semantic discrepancies between samples of the same class without explicitly modeling uncertainty. (b) The self-reflective similarity metric can perceive sample uncertainty, thereby adaptively weakening semantic discrepancies between heterogeneous samples based on this uncertainty. This enhances the robustness and reliability of similarity judgements.

rely solely on cosine similarity, SSM imposes weaker constraints on pairs with higher uncertainty. This preserves the necessary flexibility for similarity judgements, as illustrated in Figure 4. When two sample pairs exhibit identical semantic discrepancy, SSM more effectively distinguishes those with lower uncertainty. Conversely, when uncertainty significantly exceeds semantic discrepancy, SSM outputs a value of near-zero semantic discrepancy, thereby preventing unreliable labels from adversely impacting model training.

### 3.4. Self-reflective Hash Learning

As a general similarity metric scheme, RSH can be seamlessly integrated into various objective functions. Consider a training objective defined as $J(z, L; C)$, where $z$ denotes features, $L$ indicates labels, and $C$ is typically a similarity metric method. Accordingly, the training objective of RSH can be formulated as $J(z_{Sr}, L; C_{Sr})$. During inference, the cosine similarity between semantic features can be used directly as the similarity metric, while the uncertainty feature can be used optionally to calculate similarity uncertainty. Thus, compared to the original method, RSH introduces negligible computational overhead. Taking a triplet loss as an example, the training objective combined with RSH can be defined as follows:

$$J^{Sr}(z_{Sr}, L; C_{Sr}) = \sum_{a,p,n} [C_{Sr}(a,n) - C_{Sr}(a,p) + \delta]_+ \tag{10}$$

where $C_{Sr}$ denotes SSM, $\{a, p, n\}$ represents all triplet samples, $[\cdot]_+ = \max(\cdot, 0)$, and $\delta$ is a predefined interval. To avoid interference from noisy samples, this paper employs a semi-hard negative sampling strategy for sample selection (Schroff et al., 2015), which is formulated as follows:

$$n_{a,p}^* = \arg \min_{n: C_{Sr}(a,n) < C_{Sr}(a,p)} C_{Sr}(a,n) \tag{11}$$

## 3.5. Gradient Analysis

This section uses gradient analysis to analyze the impact of SSM on hash learning. Intuitively, SSM introduces uncertainty into similarity calculations to adaptively weaken semantic discrepancy, thereby reducing the influence of samples with high uncertainty on parameter updating. This effect can be measured by the magnitude of the gradients on the model parameters. First, consider the traditional training objective $J = J(z, L; C)$ using cosine similarity. According to the chain rule, its gradient with respect to the model parameters $\theta$ decomposes as follows:

$$\frac{\partial J}{\partial \theta} = \frac{\partial J}{\partial h} \cdot \frac{\partial h}{\partial \theta} = \frac{\partial J}{\partial C} \cdot \frac{\partial C}{\partial h} \cdot \frac{\partial h}{\partial \theta} \quad (12)$$

Similarly, for the training objective using SSM, the gradient of $J^{Sr} = J(z_{Sr}, L; C_{Sr})$ can be formulated as follows:

$$\frac{\partial J^{Sr}}{\partial \theta} = \frac{\partial J^{Sr}}{\partial C_{Sr}} \cdot \frac{\partial C_{Sr}}{\partial h} \cdot \frac{\partial h}{\partial \theta} = \frac{\partial J^{Sr}}{\partial C_{Sr}} \cdot \frac{\partial C_{Sr}}{\partial C} \cdot \frac{\partial C}{\partial h} \cdot \frac{\partial h}{\partial \theta} \quad (13)$$

In equations (12) and (13), the partial derivative term $\frac{\partial h}{\partial \theta}$ is determined solely by the feature encoder and is independent of the similarity modeling. Since RSH merely replaces $C$ with $C_{Sr}$ in the original training objective, the gradient form of the loss function with respect to the similarity input remains consistent. Therefore, we have $\frac{\partial J}{\partial C} = \frac{\partial J^{Sr}}{\partial C_{Sr}}$. Thus, the parameter gradient for RSH can be expressed as follows:

$$\frac{\partial J^{Sr}}{\partial \theta} = \frac{\partial J}{\partial \theta} \cdot \frac{\partial C_{Sr}}{\partial C} = \frac{\partial J}{\partial \theta} \cdot H(C, \beta), \quad (14)$$

where

$$H(C, \beta) := \frac{\partial C_{Sr}}{\partial C} = \frac{\partial(1 - C \cdot e^{-\frac{1}{\tau}\widetilde{\beta}})}{\partial C} = -e^{-\frac{1}{\tau}\widetilde{\beta}} \quad (15)$$

Gradient analysis shows that samples with high uncertainty have a smaller impact on the gradient, thereby mitigating their negative impact on hash learning. Ultimately, the model prioritises reliable samples, enhancing robustness against noise or semantic ambiguity while preserving the discriminative ability of the learned hash codes.

## 4. Experiment

### 4.1. Datasets

**MIRFlickr-25K:** consists of 24,581 image-text pairs from the Flickr website (Huiskes & Lew, 2008). The dataset involves 24 categories, each of which is annotated with at least one annotation from each class.

**MS COCO:** consists of 123,287 image–text pairs annotated with 81 categories (Lin et al., 2014). In our experiments, we merge the validation set and the training sets.

**IAPR TC-12:** consists of 20,000 natural images collected worldwide, including photographs of people, landscapes, etc., which contain 255 different categories in total (Huo et al., 2024b). Each image has the corresponding English caption.

### 4.2. Baselines and Implementation

To validate the superiority and robustness of the proposed RSH method, this paper compares it with nine representative CMH methods, including DHaPH (Huo et al., 2024b), DNPH (Huo et al., 2024a), DSTH (Qin et al., 2026), NRCH (Wang et al., 2024), RSHNL (Pu et al., 2025b), NRGH (Wang et al., 2026), DECH (Li et al., 2025), DPBE (Cheng et al., 2025), and DUaPH (Han et al., 2026). Of these methods, NRCH, RSHNL, and NRGH are robust models that have been specifically designed for noisy scenarios, whereas DECH, DPBE, and DUaPH enhance the reliability of the retrieval results by using uncertainty modeling mechanisms. To ensure a fair comparison, all of the methods use the same backbone to extract features, and they are evaluated adhering to a unified data classification standard. The parameters for each method are configured according to the specifications published in their original papers.

In the experiment, two cross-modal retrieval tasks were conducted: image-to-text ($I \rightarrow T$) and text-to-image ($T \rightarrow I$). For the RSH model, we set $\gamma = 2$ and $\tau = 9$. The proposed method was implemented using the PyTorch 2.3.0 framework on NVIDIA RTX 4090 GPUs. Unless otherwise specified, all experiments involved randomly selecting 10,000 sample pairs for the training set. 5,000 pairs for the query set, and the remainder for the retrieval database.

### 4.3. Evaluation Metric

This paper uses Mean Average Precision over all retrieved results (MAP@ALL) as the main metric for evaluating retrieval performance, which simultaneously measures the precision of results and the quality of ranking (Sharma et al., 2012). To evaluate the model's robustness under noisy labels, following prior work (Wang et al., 2026; 2024), we introduce symmetric label noise at rates of 20%, 50%, and 80% into the training set. Furthermore, this paper employs the Mean Corruption Error (MCE) to assess the extent of performance degradation in retrieval scenarios involving data corruption (Weiss & Tonella, 2022).

### 4.4. Comparison with Baselines

Table 1 presents the average MAP scores of all baselines across three datasets, from which we can draw the following conclusions: (1) The performance of common methods deteriorates significantly as the noise rate increases. Although some noise-resistant methods enhance model robustness to some extent, they struggle to maintain effective performance under high noise rates. Furthermore, while existing

*Table 1.* Performance comparison of average MAP scores of I2T and T2I tasks on the three datasets under different noise ratios. **Bold** font indicates the best results.

| Ratio | Methods | MIRFLICKR-25K | | | MS-COCO | | | IAPR TC-12 | | |
|---|---|---|---|---|---|---|---|---|---|---|
| | | 32bits | 64bits | 128bits | 32bits | 64bits | 128bits | 32bits | 64bits | 128bits |
| 0% | DHaPH (TKDE'24) | 0.8233 | 0.8251 | 0.8284 | 0.7230 | 0.7342 | 0.7382 | 0.6276 | 0.6390 | 0.6407 |
| | DNPH (TOMM'24) | 0.8061 | 0.8110 | 0.8297 | 0.6961 | 0.7341 | 0.7274 | 0.5049 | 0.5694 | 0.6320 |
| | DSTH (TMM'26) | 0.8308 | 0.8435 | 0.8559 | 0.7295 | 0.7373 | 0.7424 | 0.6474 | 0.6910 | 0.7232 |
| | NRCH (MM'24) | 0.8114 | 0.8122 | 0.8095 | 0.7263 | 0.7290 | 0.7365 | 0.5832 | 0.5891 | 0.6026 |
| | RSHNL (AAAI'25) | 0.8066 | 0.8128 | 0.8214 | 0.7430 | 0.7714 | 0.7774 | 0.5855 | 0.6354 | 0.6516 |
| | NRGH (TOMM'26) | 0.8066 | 0.8131 | 0.8126 | 0.7346 | 0.7473 | 0.7534 | 0.5894 | 0.5992 | 0.6259 |
| | DECH (AAAI'25) | 0.8277 | 0.8293 | 0.8380 | 0.6481 | 0.6821 | 0.6882 | 0.6246 | 0.6598 | 0.6727 |
| | DPBE (MM'25) | 0.8277 | 0.8434 | 0.8498 | 0.7279 | 0.7424 | 0.7565 | 0.6462 | 0.6901 | 0.7217 |
| | DUaPH (TOMM'26) | 0.8310 | 0.8431 | 0.8507 | 0.7427 | 0.7725 | **0.7981** | 0.6487 | 0.6911 | **0.7283** |
| | **RSH (Ours)** | **0.8323** | **0.8471** | **0.8581** | **0.7440** | **0.7726** | 0.7796 | **0.6492** | **0.6927** | 0.7085 |
| 20% | DHaPH (TKDE'24) | 0.7832 | 0.7963 | 0.8015 | 0.6904 | 0.7105 | 0.7354 | 0.6304 | 0.6585 | 0.6836 |
| | DNPH (TOMM'24) | 0.8091 | 0.8234 | 0.8264 | 0.7058 | 0.7290 | 0.7182 | 0.4878 | 0.5713 | 0.6230 |
| | DSTH (TMM'26) | 0.8303 | 0.8361 | 0.8384 | 0.7251 | 0.7297 | 0.7326 | 0.6440 | 0.6830 | 0.7042 |
| | NRCH (MM'24) | 0.7959 | 0.7902 | 0.7861 | 0.7133 | 0.7145 | 0.7238 | 0.5742 | 0.5866 | 0.5966 |
| | RSHNL (AAAI'25) | 0.8018 | 0.8118 | 0.8146 | 0.7343 | **0.7603** | 0.7617 | 0.5846 | 0.6321 | 0.6492 |
| | NRGH (TOMM'26) | 0.7797 | 0.7815 | 0.7669 | 0.6979 | 0.7011 | 0.7083 | 0.5590 | 0.5690 | 0.5823 |
| | DECH (AAAI'25) | 0.8244 | 0.8377 | 0.8405 | 0.6367 | 0.6757 | 0.6926 | 0.6404 | 0.6700 | 0.6760 |
| | DPBE (MM'25) | 0.8032 | 0.8207 | 0.8402 | 0.6734 | 0.6800 | 0.6975 | 0.6251 | 0.6603 | 0.7003 |
| | DUaPH (TOMM'26) | 0.8230 | 0.8319 | 0.8407 | 0.7236 | 0.7326 | 0.7533 | 0.6356 | 0.6808 | 0.7024 |
| | **RSH (Ours)** | **0.8305** | **0.8380** | **0.8458** | **0.7391** | 0.7567 | **0.7638** | **0.6468** | **0.6830** | **0.7044** |
| 50% | DHaPH (TKDE'24) | 0.7582 | 0.7638 | 0.7777 | 0.6326 | 0.6638 | 0.6819 | 0.6074 | 0.6318 | 0.6434 |
| | DNPH (TOMM'24) | 0.7985 | 0.8133 | 0.8228 | 0.6902 | 0.7121 | 0.6915 | 0.5094 | 0.5620 | 0.6216 |
| | DSTH (TMM'26) | 0.8134 | 0.8321 | 0.8390 | 0.6904 | 0.7324 | 0.7469 | 0.6442 | 0.6808 | 0.7030 |
| | NRCH (MM'24) | 0.7803 | 0.7792 | 0.7755 | 0.6878 | 0.6966 | 0.7028 | 0.5553 | 0.5570 | 0.5740 |
| | RSHNL (AAAI'25) | 0.7978 | 0.8061 | 0.8065 | **0.7227** | 0.7330 | 0.7514 | 0.5943 | 0.6350 | 0.6525 |
| | NRGH (TOMM'26) | 0.7703 | 0.7655 | 0.7538 | 0.6645 | 0.6726 | 0.6775 | 0.5432 | 0.5472 | 0.5621 |
| | DECH (AAAI'25) | 0.7888 | 0.8242 | 0.8384 | 0.5939 | 0.6612 | 0.6943 | 0.6259 | 0.6653 | 0.6811 |
| | DPBE (MM'25) | 0.7660 | 0.7872 | 0.8059 | 0.6302 | 0.6290 | 0.6355 | 0.5969 | 0.6336 | 0.6767 |
| | DUaPH (TOMM'26) | 0.7941 | 0.7991 | 0.8101 | 0.6731 | 0.7039 | 0.7126 | 0.6059 | 0.6467 | 0.6680 |
| | **RSH (Ours)** | **0.8177** | **0.8338** | **0.8464** | 0.7107 | **0.7370** | **0.7605** | **0.6455** | **0.6832** | **0.7048** |
| 80% | DHaPH (TKDE'24) | 0.7375 | 0.7416 | 0.7536 | 0.6025 | 0.6185 | 0.6585 | 0.5848 | 0.6019 | 0.6292 |
| | DNPH (TOMM'24) | 0.7965 | 0.8127 | 0.8121 | 0.6511 | 0.6665 | 0.6588 | 0.5039 | 0.5792 | 0.6201 |
| | DSTH (TMM'26) | 0.7954 | 0.8014 | 0.8094 | 0.6590 | 0.6838 | 0.7035 | 0.6375 | 0.6645 | 0.6834 |
| | NRCH (MM'24) | 0.7748 | 0.7715 | 0.7648 | 0.6742 | 0.6787 | 0.6811 | 0.5459 | 0.5502 | 0.5549 |
| | RSHNL (AAAI'25) | 0.7895 | 0.7995 | 0.7993 | 0.7049 | 0.7213 | 0.7465 | 0.5943 | 0.6299 | 0.6483 |
| | NRGH (TOMM'26) | 0.7564 | 0.7552 | 0.7451 | 0.6456 | 0.6579 | 0.6569 | 0.5346 | 0.5461 | 0.5480 |
| | DECH (AAAI'25) | 0.7444 | 0.7857 | 0.7986 | 0.5742 | 0.6202 | 0.6677 | 0.6030 | 0.6586 | 0.6749 |
| | DPBE (MM'25) | 0.7227 | 0.7552 | 0.7731 | 0.5671 | 0.5727 | 0.5806 | 0.5799 | 0.6217 | 0.6580 |
| | DUaPH (TOMM'26) | 0.7749 | 0.7761 | 0.7813 | 0.6523 | 0.6873 | 0.6921 | 0.5782 | 0.6245 | 0.6554 |
| | **RSH (Ours)** | **0.7977** | **0.8131** | **0.8140** | **0.7065** | **0.7221** | **0.7559** | **0.6402** | **0.6819** | **0.7063** |

uncertainty modeling approaches can capture sample uncertainty, they still incorporate high-uncertainty samples into similarity calculations and parameter updates, resulting in diminished performance. (2) The proposed RSH method achieves outstanding performance across various noise rates. This demonstrates that RSH effectively mitigates the interference of high-uncertainty samples during model training by adaptively weakening semantic gaps based on sample uncertainty levels. This significantly enhances the discriminability and robustness of hash codes. (3) Even under the most challenging experimental settings (e.g., noise rate is 80% and code length is 128 bits), RSH outperforms the optimal baseline methods by 3.27%, 6.38%, and 5.09% across three datasets. This result fully validates the effectiveness and superiority of RSH in resisting noisy label interference.

Furthermore, as illustrated in Figure 5, the bullet diagram shows that integrating RSH into the DHaPH and NRCH methods by replacing their original similarity metric significantly improves their retrieval performance and robustness under various noise ratios. This validates the effectiveness and advantage of RSH as a universal and robust similarity metric scheme.

### 4.5. Parametric Analysis

$\gamma$ determines self-reflective bias, while $\tau$ controls the intensity affecting similarity calculations. They jointly influence the performance of the proposed methods. As shown in Figure 6 (a), taking appropriate caution helps to suppress interference from noisy labels. Figure 6 (b) shows that a moderate degree of weakening improves retrieval perfor-

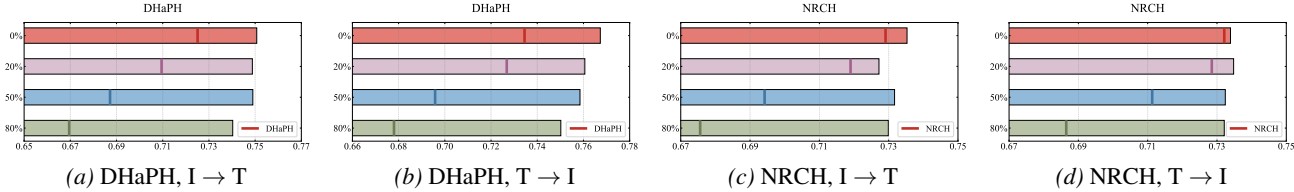

*Figure 5.* Decoupling results on MS-COCO 128 bits. The target markers indicate the baseline, and the bars correspond to adding the RSH.

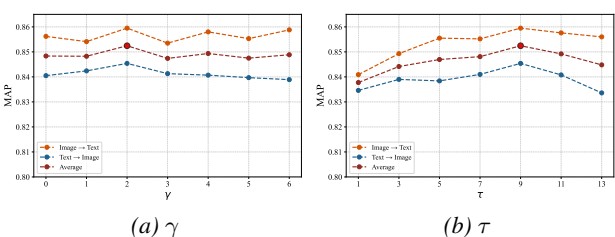

*Figure 6.* Parameter analysis for $\gamma$ and $\tau$ on MIRFLICKR-25K 128 bits under 20% noise.

*Table 2.* Ablation study results on the three datasets with 128 bits under different noisy ratios.

| Method | Image → Text | | | Text → Image | | |
|---|---|---|---|---|---|---|
| | 20% | 50% | 80% | 20% | 50% | 80% |
| MIRFLICKR-25K | | | | | | |
| RSH-$N$ | 0.8406 | 0.8198 | 0.7807 | 0.8212 | 0.8056 | 0.7789 |
| RSH-$U$ | 0.8397 | 0.8130 | 0.7805 | 0.8361 | 0.8223 | 0.7944 |
| RSH-$O$ | 0.8387 | 0.8141 | 0.7889 | 0.8375 | 0.8252 | 0.7901 |
| RSH | **0.8534** | **0.8596** | **0.8170** | **0.8382** | **0.8332** | **0.8109** |
| MS-COCO | | | | | | |
| RSH-$N$ | 0.7327 | 0.7099 | 0.6797 | 0.7386 | 0.7191 | 0.6821 |
| RSH-$U$ | 0.7383 | 0.7178 | 0.6858 | 0.7412 | 0.7235 | 0.7094 |
| RSH-$O$ | 0.7394 | 0.7126 | 0.6801 | 0.7487 | 0.7131 | 0.6947 |
| RSH | **0.7629** | **0.7608** | **0.7560** | **0.7647** | **0.7602** | **0.7558** |
| IAPR TC-12 | | | | | | |
| RSH-$N$ | 0.6850 | 0.6797 | 0.6606 | 0.6813 | 0.6693 | 0.6501 |
| RSH-$U$ | 0.6913 | 0.6814 | 0.6723 | 0.6893 | 0.6747 | 0.6733 |
| RSH-$O$ | 0.6907 | 0.6757 | 0.6523 | 0.6922 | 0.6784 | 0.6610 |
| RSH | **0.7080** | **0.7086** | **0.7077** | **0.7004** | **0.7010** | **0.7049** |

*Table 3.* Comparison of the computational efficiency of RSH and other methods with 64 bits on MS-COCO under 20% noise.

| Methods | FLOPs | Params | Train-time | Encode-time |
|---|---|---|---|---|
| DSTH | 5.579G | 151.82M | 1.31h | 1.98s |
| NRCH | 5.578G | 151.29M | 0.52h | **1.54s** |
| RSHNL | 5.579G | 151.59M | 0.63h | 2.74s |
| DPBE | 5.579G | 152.53M | 0.68h | 2.83s |
| DUaPH | 5.578G | 151.59M | 2.51h | 4.02s |
| **RSH** | **5.578G** | **151.29M** | **0.51h** | 1.68s |

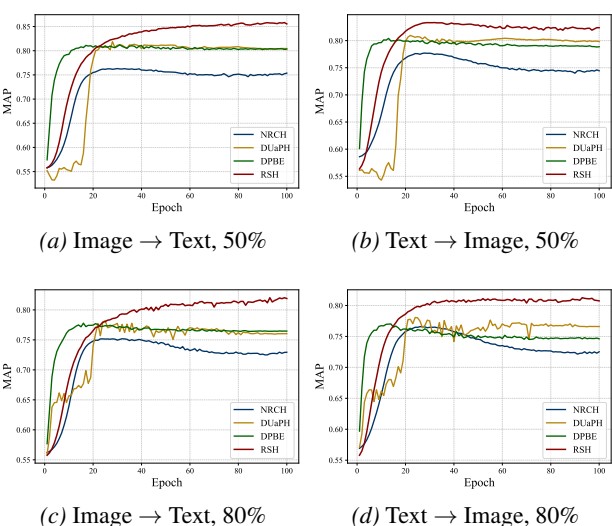

*(a)* Image → Text, 50%    *(b)* Text → Image, 50%

*(c)* Image → Text, 80%    *(d)* Text → Image, 80%

*Figure 7.* Experiment results of the MAP scores versus epochs on MIRFLICKR-25K 128 bits under 50% and 80% noise.

mance. Furthermore, the experimental results indicate that performance deteriorates as $\gamma$ and $\tau$ increase beyond a certain threshold, with excessively large values of $\gamma$ and $\tau$ proving detrimental to similarity calculations.

### 4.6. Ablation Study

To evaluate the effectiveness of each component module within RSH, we designed three variants for comparative analysis: RSH-$N$: removes the self-reflective similarity metric and solely uses cosine similarity for the metric. RSH-$U$: replaces $\|u^{\mathcal{I}}+u^{\mathcal{T}}\|_2$ with $\|u^{\mathcal{I}}\|_2+\|u^{\mathcal{T}}\|_2$ to compute uncertainty similarity, thereby ignoring uncertainty correlations

between sample pairs. RSH-$O$: constructs an alternative similarity metric that imposes greater semantic divergence on high-uncertainty sample pairs to weaken similarity judgments, defined as follows:

$$C_{Sr}(\mathcal{I},\mathcal{T}) = (1 - C(\mathcal{I},\mathcal{T})) \cdot e^{-\frac{1}{\tau}\widetilde{\beta}(\mathcal{I},\mathcal{T})} \quad (16)$$

The results presented in Table 2 yield the following conclusions: (1) RSH-$N$ struggles to suppress interference from noisy labels due to its inability to perceive uncertainty, resulting in a significant performance decline. (2) RSH-$U$ fails to capture uncertainty relationships at the sample pair level, yielding sub-optimal results. (3) RSH-$O$ attempts to amplify semantic discrepancies to mitigate the impact

*Table 4.* Performance comparison of MAP and MCE under different degradations on MIRFLICKR-25K 128 bits with 50% noise.

| Task | Method | 50% | MAP | | | MCE | | |
|---|---|---|---|---|---|---|---|---|
| | | | Gaussian Noise | Defocus Blur | JPEG Compression | Gaussian Noise | Defocus Blur | JPEG Compression |
| I→T | DNPH | 0.8289 | 0.7544 | 0.7162 | 0.7529 | 0.5860 | 0.6713 | 0.5734 |
| | DHaPH | 0.7889 | 0.7268 | 0.6992 | 0.7267 | 0.8846 | 0.9943 | 0.8974 |
| | NRCH | 0.7654 | 0.6988 | 0.6783 | 0.6973 | 0.6803 | 0.7485 | 0.6752 |
| | **RSH** | **0.8596** | **0.7743** | **0.7392** | **0.7645** | **0.5428** | **0.6124** | **0.5540** |
| T→I | DNPH | 0.8166 | 0.7644 | 0.7314 | 0.7606 | 0.6235 | 0.7172 | 0.5816 |
| | DHaPH | 0.7665 | 0.7067 | 0.6860 | 0.7042 | 0.9378 | 1.0113 | 0.9368 |
| | NRCH | 0.7855 | 0.7233 | 0.7007 | 0.7279 | 0.6134 | 0.6834 | 0.6098 |
| | **RSH** | **0.8332** | **0.7885** | **0.7450** | **0.7743** | **0.5127** | **0.5727** | **0.5032** |

of high-uncertainty samples, leading to a decline in performance. This validates the core design principle of RSH: adaptively weakening semantic discrepancies based on sample uncertainty to enhance model robustness.

### 4.7. Computational Efficiency

To evaluate the computational efficiency of the proposed method, we compared it with baseline approaches over 100 training epochs in terms of FLOPs, the number of parameters, and training and encoding time, as shown in Table 3. As all methods use the same backbone, their model sizes and FLOPs are comparable. Furthermore, compared to the latest baselines or those employing probability modeling for uncertainty, RSH demonstrates superior practicality.

### 4.8. Robust Analysis

To intuitively assess the robustness of RSH, we plotted the MAP scores against the number of epochs. As shown in Figure 7, the baselines initially indicate improved performance during training, but then decline significantly due to interference from noisy labels. In contrast, RSH shows sustained improvement in performance during the initial training stages and remains stable in subsequent epochs. This validates the advantage of RSH in guiding the learning of robust hash codes.

### 4.9. MCE Result

To validate the robustness of RSH further under incomplete and degraded sample conditions, we conducted degradation experiments during the validation phase. For the image modality, we applied Gaussian noise, defocus blur, and JPEG compression at a moderate level, as detailed in prior robustness studies (Michaelis et al., 2019). For the text modality, we simulated semantic incompleteness by randomly removing 30% of words (Weiss & Tonella, 2022). As shown in Table 4, while all methods exhibited performance degradation under degraded conditions, RSH achieved the lowest MCE and maintained relatively stable retrieval performance. RSH demonstrated a particularly significant advantage in blurred and compressed scenarios, suggesting

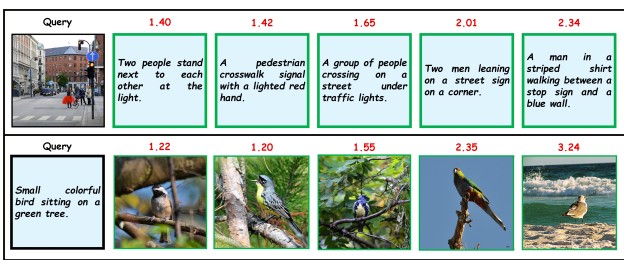

*Figure 8.* Visualization of the top-5 retrieved samples of RSH on MS-COCO 64 bits under 50% noise. Red figure denotes uncertainty for retrieval samples.

that its self-reflective similarity metric effectively mitigates the adverse effects of semantic degradation and enhances the model's robustness.

### 4.10. Visualization

Figure 8 shows examples of RSH retrieval results on the MS-COCO dataset, with uncertainty values labeled at the top of each retrieved sample. The uncertainty value is defined as the L2-norm of its uncertainty feature vector. Generally, higher uncertainty indicates lower retrieval reliability. By recognizing sample uncertainty and adapting to mitigate semantic discrepancies, RSH can effectively suppress interference from noisy labels and retrieve positive samples accurately, even in noisy scenarios. This intuitively demonstrates the robustness of the proposed method.

## 5. Conclusion

This paper proposes a Robust Self-reflective Hashing (RSH) framework that explicitly incorporates uncertainty into similarity modeling and prudently adjusts semantic differences, thereby effectively mitigating the interference of noisy labels or semantic ambiguity. Specifically, we design a dual feature representation mechanism that simultaneously captures both the semantic information and the uncertainty level of samples. Furthermore, we propose a self-reflective similarity metric that jointly models semantic discrepancies and uncertainty, enabling the model to adaptively weaken semantic constraints based on uncertainty levels. Extensive

experiments on benchmark datasets demonstrate that the proposed RSH method achieves significant improvements in both retrieval performance and robustness.

## Acknowledgements

This work was supported by the National Natural Science Foundation of China (No.62302338, No. 62472390), Shandong Provincial Natural Science Foundation (No.ZR2025MS1067).

## Impact Statement

This paper presents work whose goal is to advance the field of Machine Learning. There are many potential societal consequences of our work, none of which we feel must be specifically highlighted here. Our method aims to improve the robustness of cross-modal retrieval systems under noisy supervision and semantic ambiguity, which may benefit large-scale multimedia retrieval applications in real-world scenarios. Furthermore, the proposed method is plug-and-play, making it practical for deployment in efficient retrieval systems.

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
