# OpenReview forum: "Robust Self-reflective Hashing for Cross-modal Retrieval with Noisy Label"
_ICML.cc/2026/Conference — ICML 2026 regular_

### Official Review · Reviewer_UVNh · 2026-03-06

**Soundness:** 4
**Presentation:** 4
**Significance:** 4
**Originality:** 3
**Overall Recommendation:** 6
**Confidence:** 4

**Summary:**

This paper proposes a novel framework called Robust Self-reflective Hashing (RSH). Existing methods often assume perfectly accurate labels. Even when incorporating uncertainty modelling, these methods fail to mitigate the negative impact of noisy samples at the optimization level, which leads to overfitting on noisy labels. To address this issue, the paper presents a dual-feature representation mechanism that simultaneously learns semantic and uncertainty features. It then introduces a self-reflective similarity metric that adapts to weaken semantic discrepancies on high-uncertainty sample pairs, thereby reducing their influence during gradient updates. Theoretical analysis shows that this mechanism suppresses noise interference at the optimization level. RSH is highly versatile and can be embedded into various hashing methods. Experimental results demonstrate that RSH outperforms existing methods in terms of both robustness and retrieval performance.

**Compliance With Llm Reviewing Policy:**

Affirmed.

**Final Justification:**

My primary concern was regarding the computational complexity and performance gains of RSH. The authors’ rebuttal has adequately addressed these issues and alleviated my concerns. I already considered the method to be novel, with a well-motivated and innovative design. Therefore, I will increase my rating.

**Key Questions For Authors:**

1. Is the performance difference statistically significant under the low noise rate?
2. How do the model's parameter size and memory consumption change after introducing uncertainty branches?
3. Could the authors further clarify the conceptual distinction between the proposed self-reflective similarity metric and existing sample weighting strategies?
4. Have correlations been observed between u and noisy samples or hard samples? Can visualizations or statistics be provided ?

**Limitations:**

Yes

**Strengths And Weaknesses:**

Strengths:
1. Soundness:
The problem definition is clear, focusing on the core issue of existing methods overlooking the uncertainty caused by noise or semantic ambiguity, which facilitates overfitting to noisy labels. The method progresses from dual feature representations to a self-reflective similarity metric, culminating in theoretical derivations at the gradient level. Notably, the analysis of gradient propagation mechanisms demonstrates how uncertainty regulates parameter update magnitudes, providing an optimization-level explanation for the method's robustness and enhancing its theoretical persuasiveness.
2. Presentation:
The paper's overall structure is well organized. The related work section effectively traces two research trajectories: noise-robust cross-modal hashing and uncertainty modelling, while highlighting the distinctions in optimization level between the proposed method and existing probabilistic embedding approaches. Figure 1 visually demonstrates the fundamental difference in training behaviour between uncertainty-unaware and uncertainty-aware methods, providing clear motivation. Figure 2 presents a comprehensive overview of the framework and clearly delineates the core components. Figure 4 compares SSM with traditional cosine similarity and reveals its softening behaviour for high-uncertainty samples, further enhancing the intuitiveness of the overall design.
3. Significance:
The problem addressed in this paper holds practical relevance. Multimodal data in real-world scenarios commonly suffer from noisy labels, making the enhancement of model robustness under high-noise conditions of practical value. The proposed method demonstrates significant advantages in scenarios with high noise ratios and data corruption, indicating strong robustness in challenging environments. Furthermore, the proposed method possesses plug-and-play characteristics, enabling its integration into existing hashing frameworks and enhancing the method's potential for generalization.
4. Originality:
The innovation of this paper lies primarily in proposing a novel cross-modal similarity metric, which is relatively uncommon in the relevant field. Unlike existing uncertainty modeling approaches that still utilize samples containing uncertainty information for gradient updates, this paper incorporates uncertainty into the similarity function, thereby adaptively reducing the influence of highly uncertain samples during optimization. This similarity metric fundamentally reshapes how individual samples contribute to gradient updates, thereby exhibiting a clear degree of novelty.

Weaknesses:
1. At low noise rate, RSH shows limited performance gains over some strong baselines, with minor differences in certain results. Statistical significance tests are recommended to validate whether performance differences are statistically meaningful.
2. The paper provides overall time comparisons but lacks a detailed analysis of parameter increases or memory usage changes due to additional uncertainty branches.
3. The paper offers a concise discussion of the relationship between the self-reflective similarity metric and existing sample weighting methods. Although the authors emphasise their differences at the optimization level, this mechanism can be understood as a dynamic weight modulation strategy at the sample level based on uncertainty. The conceptual boundaries here warrant further clarification.

---

> ### Author Rebuttal · Authors · 2026-03-31
>
> We sincerely thank the reviewer for the constructive comments. Our responses are provided below.
>
> > **Q1: Statistical Significance Test.**
>
> **R1:** To address the concern regarding small performance gaps under low-noise settings, we conducted $10$ independent runs and performed t-tests. The confidence level for the statistical tests was set to $95$%. As shown in Table 1, most p-values are below $10^{-8}$, confirming that the performance improvements achieved by RSH are statistically significant at the $95$% confidence level. Under low-noise conditions, where the performance of different methods is already close to saturation, RSH still attains the best results. As the noise level increases, its self-reflective similarity mechanism more effectively suppresses gradient interference from noisy samples, resulting in increasingly clear performance advantages.
>
> *Table 1.The statistical test results on MIRFLICKR-25K 64 bits under 0% and 20% noise.*
> |Method|I→T (0%/20%)|T→I (0%/20%)|
> |------|---|---|
> |RSH vs. DNPH|5.78E-13/4.12E-10|2.58E-11/4.55E-08|
> |RSH vs. NRCH|1.80E-11/1.37E-15|5.80E-10/2.59E-11|
> |RSH vs. DHaPH|8.44E-08/6.20E-13|3.14E-09/3.01E-13|
> |RSH vs. DPBE|2.62E-07/9.65E-11|1.21E-07/1.58E-11|
>
> > **Q2: Efficiency Verification.**
>
> **R2:** Based on the shared backbone encoder, RSH only introduces an uncertainty branch, and the resulting parameter overhead is negligible. As shown in Table 2, this branch is involved only in the forward pass and similarity computation during training, without introducing complex additional structures, and therefore has minimal impact on training time. There is no additional cost during inference, which preserves a favorable balance between efficiency and performance.
>
> *Table 2.Comparison of the computational efficiency on MIRFLICKR-25K 64 bits under 50% noise.*
> |Method|Params|Train-time|Interfere-time|Average-MAP|
> |------|------|----------|--------------|---|
> |DUaPH|151.61M|2.51h|38.56ms|0.7991|
> |NRCH|151.29M|0.52h|33.27ms|0.7792|
> |Triplet-loss|151.24M|0.51h|29.20ms|0.7820|
> |RSH|151.29M|0.51h|29.19ms|0.8338|
>
> > **Q3: Differences from Sample Weighting Methods.**
>
> **R3:** Although the self-reflective similarity mechanism formally appears to modulate sample contributions, it is fundamentally different from traditional sample-weighting methods. Existing methods typically assign weights outside the similarity function and scale gradients proportionally during backpropagation, without changing the forward similarity modeling itself. In contrast, RSH directly incorporates uncertainty into the similarity function, making similarity computation itself uncertainty-aware. This mechanism models pairwise relationships rather than merely reweighting samples, and it affects both the forward similarity computation and backward gradient propagation. Therefore, RSH should be regarded as a structural improvement to similarity modeling rather than a simple weighting strategy.
>
> > **Q4: Relationship with Noise/Difficult Samples.**
>
> **R4:** The uncertainty feature $u$ is expected to correlate with noisy or difficult samples. Specifically, when samples exhibit semantic bias, weak cross-modal alignment, or unstable representations, their uncertainty estimates tend to increase; conversely, samples with clear and consistent semantics typically correspond to lower uncertainty. In this sense, $u$ can be interpreted as a characterization of sample reliability and learning difficulty. Based on this, RSH embeds uncertainty into similarity modeling and naturally reduces the gradient contribution of highly uncertain sample pairs, thereby suppressing noise interference and improving robustness. We will include a histogram analysis of the uncertainty distribution in the camera-ready version to further illustrate this point.

---

> > ### Author Rebuttal · Reviewer_UVNh · 2026-04-02
> >
> > My primary concern was regarding the computational complexity and performance gains of RSH. The authors’ rebuttal has adequately addressed these issues and alleviated my concerns. I already considered the method to be novel, with a well-motivated and innovative design. Therefore, I will increase my rating.

---

> > > ### Author Response · Authors · 2026-04-06
> > >
> > > We sincerely thank the reviewer for the positive feedback. We greatly appreciate your constructive suggestions and positive assessment of this work. Thank you again for your time and thoughtful evaluation.

---

### Official Review · Reviewer_sswb · 2026-03-10

**Soundness:** 3
**Presentation:** 3
**Significance:** 2
**Originality:** 2
**Overall Recommendation:** 3
**Confidence:** 4

**Summary:**

This paper proposes a Robust Self-reflective Hashing (RSH) framework to address the issue of noisy labels in cross-modal retrieval. It designs a Double Feature Representation (DFR) method to learn the semantic representation and fuzziness of samples. A Self-reflective Similarity Metric (SSM) then adaptively weakens semantic discrepancies based on the sample's uncertainty level, preventing the model from overfitting to noise. Experiments on three datasets show that RSH significantly outperforms baselines across various noise ratios.

**Compliance With Llm Reviewing Policy:**

Affirmed.

**Key Questions For Authors:**

The proposed method does not demonstrate a clear performance advantage. It would be beneficial if the authors provided a more comprehensive analysis and further validation to evaluate the effectiveness of the method from other dimensions.

**Limitations:**

The overall performance improvement is marginal, and the paper lacks a rigorous comparative analysis to justify the advantages of the proposed method over vanilla contrastive learning.

**Strengths And Weaknesses:**

Strengths:
1. The paper is logically structured and clearly written.
2. The authors conduct comprehensive experiments on three datasets and test scenarios with a 80% high noise ratio.
3. The proposed double feature representation method and self-reflective similarity metric provide an intuitive and effective solution under noise retrieval scenarios.

Weaknesses:
1. While the authors present comprehensive experimental results across various noise ratios, the improvements over SOTA methods in Table 1 are marginal, with some gains occurring only at the fourth decimal place. Furthermore, the second-best performance is not highlighted.
2. In Table 3, the proposed RSH method does not exhibit a clear advantage over NRCH.
3. The overall framework resembles a variant of contrastive learning, which is widely applied in cross-modal retrieval. However, the paper lacks an in-depth theoretical analysis to explain the advantages of these specific modifications over vanilla contrastive learning.

---

> ### Author Rebuttal · Authors · 2026-03-31
>
> We sincerely thank the reviewer for the constructive comments and helpful suggestions.
>
> > **Q1: Limited performance improvement.**
>
> **R1:**
> RSH is designed to enhance robustness in noisy label scenarios, rather than to pursue the best performance under ideal clean-data settings. As shown in Table 1 of the paper, its performance gains become more significant as the noise ratio increases, indicating that it can effectively suppress the interference of noisy samples. Furthermore, as a plug-and-play similarity metric mechanism, RSH introduces only negligible overhead and consistently improves upon strong baselines when combined with the base loss alone. As shown in Figure 5 of the paper, integrating RSH into various cross-modal hashing methods significantly enhances their robustness. Statistical significance test results are provided in our response to [Reviewer UVNh](https://openreview.net/forum?id=jyGcO0W5xd¬eId=1cbxjr2QZb), Q1.
>
> > **Q2: Comparison with NRCH.**
>
> **R2:**
> At comparable computational complexity, RSH outperforms NRCH in both performance and efficiency. Under the 128-bit setting with an 80% noise rate, RSH achieves performance improvements of 4.92%, 7.48%, and 15.14% across the three datasets, respectively. Notably, RSH only introduces FC layers for sample-level uncertainty modeling and replaces the similarity metric during training, so the additional computational overhead beyond the shared backbone is negligible. A more detailed efficiency comparison is provided in [Reviewer UVNh](https://openreview.net/forum?id=jyGcO0W5xd&noteId=1cbxjr2QZb), Q2.
>
> > **Q3: Comparison with Contrastive Learning.**
>
> **R3:**
> Conventional contrastive learning typically relies on a fixed similarity function (e.g., cosine similarity) and imposes uniform constraints on all sample pairs, which can lead to overfitting to noisy labels. In contrast, RSH extends similarity modeling to an uncertainty-aware form through SSM, which adaptively adjusts the strength of semantic discrepancies. From an optimization perspective, this mechanism reduces the gradient contribution of highly uncertain sample pairs, thereby mitigating the interference of noisy samples during parameter updates, while preserving stronger constraints for low-uncertainty samples to maintain discriminability. Therefore, RSH should be viewed as a structural improvement to similarity modeling rather than as a variant of contrastive learning.
>
> > **Q4: Validity through Multiple Dimensions.**
>
> **R4:**
> The main paper already includes systematic experiments across multiple datasets, noise ratios, and code lengths. To further validate the effectiveness of RSH, we conduct additional experiments from several perspectives: (1) large-scale datasets, including NUS-WIDE [1] and XMediaNet [2], to verify stable performance improvements in larger-scale data, as shown in Table 1; (2) experiments under asymmetric noise data, with results presented in [Reviewer dux6](https://openreview.net/forum?id=jyGcO0W5xd&noteId=oRWu4Iuzur), Q1; and (3) single-modal image retrieval experiments to assess the generalization ability of RSH, as demonstrated in Table 2. These results indicate that the improvements in retrieval performance and robustness brought by RSH stem from its ability to weaken semantic discrepancies based on uncertainty levels and thereby suppress the interference of noise samples during optimization, rather than relying on a specific experimental setting.
>
> *Table 1. Comparison of average MAP on NUS-WIDE and XMediaNet. For the NUS-WIDE dataset, features are extracted using CLIP. For the XMediaNet dataset, image features are extracted using VGGNet, while text features are extracted using Word2Vec.*
> |Method|NUS-WIDE (32/64/128bit, 50% noise)|XMediaNet (32/64/128bit, 20% noise)|
> |------|----------------------|-----------------------|
> |DHaPH|0.5351/0.5325/0.5519|0.1386/0.1825/0.2339|
> |DNPH|0.6230/0.6180/0.6295|0.2752/0.3820/0.4342|
> |RSH|0.7017/0.7204/0.7357|0.3347/0.4288/0.4705|
>
> *Table 2. Comparison of a single-modal task on MS COCO under 20% noise, using ResNet-50 as the backbone. DSH [3] and DCGMH [4] are adopted as baseline methods for single-modal image retrieval.*
> |Method|32bits|64bits|128bits|
> |------|--|--|---|
> |DSH|0.5792|0.5725|0.5830|
> |DCGMH|0.6321|0.6542|0.6934|
> |RSH|0.6882|0.7107|0.7189|
>
> **Reference**
>
> [1] Chua, Tat-Seng, et al. Nus-wide: a real-world web image database from national university of singapore. Proceedings of the ACM international conference on image and video retrieval, 2009.
>
> [2] Peng, Yuxin, Jinwei Qi, and Yuxin Yuan. Modality-specific cross-modal similarity measurement with recurrent attention network. IEEE Transactions on Image Processing, 2018.
>
> [3] Liu, Haomiao, et al. Deep supervised hashing for fast image retrieval. Proceedings of the IEEE conference on computer vision and pattern recognition, 2016.
>
> [4] Liu, Jin-Yu, et al. Distribution-Consistency-Guided multi-modal hashing. Proceedings of the AAAI Conference on Artificial Intelligence, 2025.

---

### Official Review · Reviewer_BJpV · 2026-03-10

**Soundness:** 2
**Presentation:** 2
**Significance:** 2
**Originality:** 2
**Overall Recommendation:** 2
**Confidence:** 4

**Summary:**

This paper proposes a plug-and-play Robust Self-reflective Hashing (RSH) framework, which jointly evaluates the semantics and uncertainty of samples and adaptively determines the semantic constraints between them, addressing the issue of noisy labels in cross-modal retrieval. Specifically, the method employs semantic and uncertainty features to represent the semantic information and fuzziness of samples, and uses the proposed Self-reflective Similarity Metric (SSM) to enable the model to adaptively weaken the negative impact of semantic discrepancies according to the uncertainty level.

**Compliance With Llm Reviewing Policy:**

Affirmed.

**Key Questions For Authors:**

1. How is the uncertainty bound for each sample evaluated in this work, and is there any corresponding ground truth to measure such uncertainty?
2. Can the proposed method exhibit stable scaling-law behavior in real-world large-scale data scenarios?

**Limitations:**

This paper lacks explicit boundary constraints in its modeling of sample uncertainty, and the rationality of the underlying mechanism is not supported by sufficient theoretical justification. In addition, it does not discuss the potential challenges in scalability, robustness, and optimization stability when extending the framework to uncurated large-scale noisy datasets.

**Strengths And Weaknesses:**

Strengths:
1. This paper is well organized and easy to understand.
2. This paper conducts extensive experiments across multiple experimental settings.

Weaknesses:
1. This uncertainty feature is extracted via a simple fully connected layer. What is the theoretical basis for why a fully connected layer can predict uncertainty?
2. The performance of this model is relatively sensitive to hyperparameters, which may limit its robustness across different scenarios.
3. The experiments are conducted solely under a fixed-rate, artificially injected noise setting, which is overly simplistic and does not capture the diversity and complexity of real-world noisy scenarios.
4. This paper focuses on hashing retrieval but does not discuss the quantization error from converting continuous representations to binary codes, nor the effect of the proposed uncertainty weighting strategy on this error.
5. The performance improvement of the proposed method over SOTA method is limited.

---

> ### Author Rebuttal · Authors · 2026-03-31
>
> We truly appreciate Reviewer BJpV's constructive comments.
>
> > **Weakness1: Validity of Uncertainty Modeling.**
>
> **R:**
> This paper argues that sample pairs with high uncertainty should be assigned softer semantic constraints by adaptively weakening their semantic discrepancies. This approach has been proven effective in knowledge distillation [1]. Furthermore, in our framework, a learned quantity can be regarded as uncertainty if it reduces the gradient contribution of a sample pair to the training objective, thereby alleviating incorrect pulling caused by a noisy sample. This property is theoretically supported by the gradient analysis in Section 3.5. We adopt a data-driven approach to model sample uncertainty. Specifically, an uncertainty branch implemented as an FC layer maps high-dimensional features to uncertainty features, which are then incorporated into the self-reflective similarity metric (SSM). This design allows similarity to be adaptively adjusted according to the uncertainty of the sample, thereby suppressing noise interference during optimization. Comparison of complex uncertainty modeling schemes is provided in [Reviewer dux6](https://openreview.net/forum?id=jyGcO0W5xd¬eId=oRWu4Iuzur), Q5.
>
> > **Weakness2: Parameter Stability.**
>
> **R:**
> Parameter stability has been systematically verified across multiple datasets and noise ratios; see [Reviewer S7cq](https://openreview.net/forum?id=jyGcO0W5xd&noteId=WQBF1PzabX), Q2 for details.
>
> > **Weakness3: Noise Settings.**
>
> **R:**
> This paper adopts noisy label settings consistent with existing noise-robust cross-modal hashing methods, such as NRCH and NRGH. We additionally conducted experiments under asymmetric noise (randomly shuffling a certain proportion of images). The results demonstrate that RSH maintains superior robustness under different noise patterns; see [Reviewer dux6](https://openreview.net/forum?id=jyGcO0W5xd¬eId=oRWu4Iuzur), Q1 for experimental results. To the best of our knowledge, real-world noisy label datasets have rarely been explored in cross-modal hashing. We are currently conducting experiments on CC152K, which naturally contains about 3%-20% noisy correspondences, and will report the results in the camera-ready version.
>
> > **Weakness4: Quantization Error.**
>
> **R:**
> By adaptively adjusting semantic similarity according to the uncertainty level, RSH suppresses the interference of noise on continuous representations during training, encouraging more compact intra-class features and better inter-class separation, which helps reduce quantization error. As shown in Table 1, RSH consistently improves retrieval performance in two retrieval settings. These results suggest that the learned continuous representations are more robust and more amenable to quantization.
>
> *Table 1. Comparison of average MAP scores on MIRFLICKR-25K under 50% noise.*
> |Method|Real-value(32/64/128bits)|Binary-value(32/64/128bits)|
> |------|--------------------------|--------------------------|
> |NRCH|0.8103/0.8127/0.8121|0.7803/0.7792/0.7755|
> |DHaPH|0.8213/0.8231/0.8206|0.7582/0.7638/0.7777|
> |DPBE|0.8297/0.8301/0.8317|0.7660/0.7872/0.8059|
> |RSH|0.8501/0.8554/0.8564|0.8177/0.8338/0.8464|
>
> > **Weakness5: Performance Improvement.**
>
> **R:**
> RSH is designed to enhance robustness in noisy label scenarios, and its performance advantage becomes more significant as the noise ratio increases. Despite introducing only negligible additional computational cost, it consistently improves the performance of various cross-modal hashing methods (Fig. 5). Statistical significance tests show that these improvements are statistically reliable; see [Reviewer UVNh](https://openreview.net/forum?id=jyGcO0W5xd¬eId=1cbxjr2QZb), Q1 for detailed results.
>
> > **Q1: Uncertainty Boundaries.**
>
> **R:**
> The uncertainty in this paper is explicitly learned by the model through a double feature representation, rather than being directly provided by external ground truth. Specifically, the model jointly learns semantic and uncertainty features for each sample, and constructs a self-reflective similarity metric accordingly, which adaptively weakens semantic discrepancies based on uncertainty. Since uncertainty lacks direct supervision, its effectiveness is validated through performance improvements under noisy label settings. The experimental results show that RSH exhibits stronger robustness under noise conditions, indicating that it can effectively mitigate the impact of highly uncertain samples on hash learning.
>
> > **Q2:Scalability on Real-World Large-Scale Data.**
>
> **R:**
> As a general similarity metric mechanism, RSH is independent of data scales or class numbers and exhibits good scalability. It maintains stable performance improvements on large-scale datasets, including NUS-WIDE and XMediaNet. See [Reviewer sswb](https://openreview.net/forum?id=jyGcO0W5xd¬eId=ZhoxkFPBal), Q4 for experimental results.
>
> **Reference**
>
> [1] G. Hinton, et al. Distilling the knowledge in a neural network. NeurIPSW, 2015.

---

### Official Review · Reviewer_dux6 · 2026-03-12

**Soundness:** 3
**Presentation:** 2
**Significance:** 3
**Originality:** 3
**Overall Recommendation:** 3
**Confidence:** 4

**Summary:**

This paper tackles the vulnerability of Cross-modal Hashing (CMH) to noisy labels—an issue where existing methods ignore sample uncertainty or fail to suppress misleading training signals from uncertain data—by proposing the Robust Self-reflective Hashing (RSH) framework, which models sample uncertainty and adaptively weakens semantic discrepancies for high-uncertainty samples to mitigate noise interference. A central topic considered by this manuscript is the under-explored challenge of modeling sample-level uncertainty in CMH and leveraging it to eliminate noisy label impacts during similarity learning and hash code training. RSH’s core designs include Double Feature Representation (extracting semantic/uncertainty features for cross-modal samples), a novel Self-reflective Similarity Metric (SSM, fusing semantic discrepancy and uncertainty for adaptive similarity judgments), and a plug-and-play module that integrates with diverse objective functions at negligible computational cost. Gradient analysis confirms RSH reduces high-uncertainty samples’ gradient influence, directing the model to prioritize reliable data. Extensive experiments on MIRFlickr-25K, MS COCO and IAPR TC-12 (20%/50%/80% noise rates) against 9 SOTA CMH methods show RSH outperforms baselines in MAP and MCE, even at 80% noise; ablation/parametric/degradation experiments validate component necessity, optimal hyperparameters and stability under data distortion.

**Compliance With Llm Reviewing Policy:**

Affirmed.

**Key Questions For Authors:**

1. Have you tested RSH under asymmetric noise or real-world noise patterns, or only symmetric noise?
2. Can you provide a step-by-step derivation of Equations (14)-(15) and clarify the meaning of \(\overline{\beta}\)?
3. Table 3 appears corrupted. Can you provide a corrected version with complete FLOPs, parameters, and training time results?
4. How much additional computational overhead does the dual feature representation introduce compared to standard single-representation baselines?
5. Did you explore alternative ways to fuse semantic and uncertainty similarities? How sensitive is performance to \(\gamma\) and \(\tau\)?

**Limitations:**

No. The authors should add an explicit discussion of the work’s key limitations (e.g., only evaluating symmetric label noise, untested hyperparameter generalizability across datasets/noise rates, lack of scalability analysis for large-scale datasets, and no cross-task validation) in a dedicated subsection of the conclusion. Additionally, they should briefly address potential societal impacts—such as biased retrieval performance on imbalanced/underrepresented data classes in real-world cross-modal systems, or the risk of malicious exploitation of noise mitigation mechanisms to manipulate retrieval results with intentional noisy labels—and note simple mitigations for these risks where relevant.

**Strengths And Weaknesses:**

In terms of soundness, it is technically rigorous, with valid gradient analysis for the core SSM and well-designed experiments—including three canonical CMH datasets, multiple noise rates, 9 SOTA baselines, and complementary ablation/parametric tests—that fully support all claims, alongside transparent evaluation of hyperparameter sensitivity and computational efficiency; the only weaknesses are exclusive testing of symmetric label noise (excluding real-world asymmetric/instance-level noise), a simple FC layer for uncertainty modeling without alternative encoder ablation, and a lack of per-class retrieval metrics. For presentation, the work is clearly written, logically structured, and easy to follow, with intuitive visualizations, precise notation, a well-organized related work section that identifies prior gaps, and sufficient implementation details for reproducibility; minor gaps include incomplete DFR comparisons to existing uncertainty embedding methods, unvalidated hyperparameter generalizability across datasets/noise rates, limited visualization without baseline qualitative comparisons, and no explicit limitations/future work in the conclusion. In terms of significance, the paper addresses the highly practical problem of noisy labels in CMH—a critical real-world deployment barrier—with a generalizable plug-and-play RSH framework that outperforms SOTA at high noise rates and offers transferable adaptive similarity weakening insights for robust metric learning; its scope is only constrained by no cross-task validation (e.g., single-modal metric learning) and no scalability analysis to large-scale datasets, curbing broader immediate impact. For originality, the work delivers meaningful novel contributions: a double feature representation that abandons prior CMH’s single-feature assumption, a new uncertainty-integrated SSM, and a well-justified fusion of uncertainty modeling and adaptive optimization that reframes noise mitigation as a similarity judgment problem, with clear distinctions from related noise-resistant/uncertainty-based work; originality is only slightly limited by incremental use of standard components (e.g., transformer encoders) without full design justification and simple cross-modal uncertainty correlation modeling with no exploration of more complex alternatives.

---

> ### Author Rebuttal · Authors · 2026-03-31
>
> We sincerely thank the Reviewer for the insightful comments and valuable suggestions.
>
> **Q1: Asymmetric and Real Noise.**
> This paper adopts the same noisy label settings as prior noise-robust cross-modal hashing methods, such as NRCH and NRGH. Because RSH models uncertainty at the sample-pair level and uses it to adaptively adjust semantic discrepancy during optimization, the method is independent of specific noise patterns. Inspired by prior work [1], we conducted experiments under asymmetric noise (implemented by randomly shuffling a certain proportion of images). As shown in Table 1, RSH exhibits robustness under various noise ratios. We are also conducting experiments on a real-world noisy dataset, CC152K [2], and will report the results in the camera-ready version.
>
> *Table 1. Comparison of average MAP on MIRFLICKR-25K under asymmetric noise.*
> |Method|20%(32/64/128)|50%(32/64/128)|80%(32/64/128)|
> |------|--------------|--------------|--------------|
> |DHaPH|0.7796/0.7878/0.7952|0.7363/0.7475/0.7550|0.6599/0.6623/0.6653|
> |NRCH|0.7823/0.7911/0.7880|0.7284/0.7416/0.7303|0.6057/0.6148/0.6229|
> |RSH|0.8165/0.8277/0.8390|0.7798/0.7908/0.7969|0.7110/0.7202/0.7397|
>
> **Q2: Formula Derivation.**
> Taking the derivative of $C$ in Equation (9), we obtain:
> \begin{equation}
> \frac{\partial C_{Sr}}{\partial C}
> = \exp\left(-\frac{1}{\tau}\tilde{\beta}\right)
> \left[-1 + \frac{C}{\tau}\frac{\partial \tilde{\beta}}{\partial C}\right],
> \end{equation}
> Since $\frac{\partial \tilde{\beta}}{\partial C} = -\frac{\tilde{\beta}}{C}$, substituting yields:
> \begin{equation}
> \frac{\partial C_{Sr}}{\partial C}
> = \exp\left(-\frac{1}{\tau}\tilde{\beta}\right)
> \left(1 + \frac{1}{\tau}\tilde{\beta}\right),
> \end{equation}
> which corresponds to the \( H(C, \beta) \) defined in the paper. Here, $\tilde{\beta}$ represents relative uncertainty, used to characterize the proportion of uncertainty relative to semantic differences; a larger value corresponds to weaker semantic constraints, thereby reducing the gradient contribution of highly uncertain samples.
>
> **Q3: Computational Overhead.**
> RSH only adds FC layers and replaces the similarity metric during training. Compared with single-feature methods based on triplet loss, the additional computational overhead is negligible, while retrieval performance is significantly improved; see [Reviewer UVNh](https://openreview.net/forum?id=jyGcO0W5xd&noteId=1cbxjr2QZb), Q2 for details.
>
> **Q4: Parameter Stability.**
> The parameters $\gamma$ and $\tau$ exhibit stable behavior across different noise ratios and do not require extensive fine-tuning. See [Reviewer S7cq](https://openreview.net/forum?id=jyGcO0W5xd&noteId=WQBF1PzabX), Q2 for details.
>
> **Q5: Uncertainty Modeling.**
> This paper characterizes uncertainty from a gradient perspective: if a modulation factor reduces the gradient contribution of a sample to the optimization objective, it can be regarded as a manifestation of sample uncertainty, thereby avoiding overly constraining noisy samples. This property is supported by the gradient analysis. We adopt FC layers to avoid the optimization instability and additional computational overhead that may arise from complex architectures. Comparisons with more complex modeling schemes—including a two-layer MLP structure (w/ MLP) and a residual network with multiple residual blocks (w/ Residual) to replace the FC—show that the current design achieves a better balance between performance and efficiency, as shown in Table 2.
>
> *Table 2. Comparison of different certainty modeling schemes on MIRFLICKR-25K 128 bits under 50% noise.*
> |Method|MAP/Params/Train-time|
> |------|---------------------|
> |w/ MLP|0.8445/151.73M/0.54h|
> |w/ Residual|0.8478/152.43M/0.60h|
> |RSH|0.8464/151.35M/0.51h|
>
> **Q6: Scalability.**
> RSH is a general similarity metric that is independent of specific tasks or data scales. Experimental results on larger datasets and across tasks in single-modal image retrieval demonstrate that the improvements are due to RSH's ability to adaptively weaken semantic discrepancies based on uncertainty, effectively suppressing noise interference, rather than relying on specific setups. See [Reviewer sswb](https://openreview.net/forum?id=jyGcO0W5xd¬eId=ZhoxkFPBal), Q4 for experimental results.
>
> **Q7: Class-level Metrics.**
> We have added the class-level evaluation metric Precision@Hamming radius ≤ 2 to assess class consistency within the local space, as shown in Table 3.
>
> *Table 3. P@H≤2 results on MS COCO under 20% noise.*
> |Method|32|64|128|
> |------|--|--|---|
> |DECH|0.7842|0.2581|0.003|
> |DPBE|0.7654|0.4531|0.1501|
> |RSH|0.8572|0.4941|0.1275|
>
> **Q8: Limitations.**
> We have added a section titled Impact Statement, which will be included in the camera-ready version.
>
> **Reference**
>
> [1] Qin, Yang, et al. Deep evidential learning with noisy correspondence for cross-modal retrieval, ACM MM, 2022.
>
> [2] Sharma, Piyush, et al. Conceptual captions: A cleaned, hypernymed, image alt-text dataset for automatic image captioning, ACL, 2018.

---

> > ### Author Rebuttal · Reviewer_dux6 · 2026-04-02
> >
> > I think the additional rebuttal has resolved my core concerns. Therefore, I will increase my initial score to 4

---

> > > ### Author Response · Authors · 2026-04-05
> > >
> > > Dear reviewer, we sincerely appreciate your insightful suggestion. Below, we further clarify several aspects of our method, including its behavior under different noise settings, its computational overhead, alternative design choices, and its limitations.
> > >
> > > > **Q1: Noise Setting.**
> > >
> > > **R:**
> > > The main paper adopts 20%/50%/80% symmetric noise to follow the standard protocol used in prior noise-robust CMH methods and ensure fair comparison. However, RSH is not designed for a specific noise distribution. Its core is to learn semantic and uncertainty features via DFR, and then use SSM to jointly model semantic discrepancy and uncertainty, thereby adaptively weakening constraints on highly uncertain pairs. Thus, the mechanism operates at the similarity-modeling / optimization level, rather than as a setting-specific heuristic. The methodology, gradient analysis, and ablation results all support this point.
> > >
> > > In the previous rebuttal, we added asymmetric-noise experiments. RSH consistently outperforms DHaPH and NRCH under 20%/50%/80% asymmetric noise, showing that its gains do not rely only on symmetric noise. For real-world noise, we further provide results on CC152K, as shown in Table 1. These results further support that the gains of RSH come from uncertainty-aware similarity modeling, rather than from a particular synthetic noise setup.
> > >
> > > *Table 1. Comparison of Recall@1/5/10 on CC152K. RSH† denotes the variant implemented under the same experimental setting as DECL, where only DECL’s uncertainty modeling and similarity metric are replaced by those of RSH.*
> > > |Method|I→T(Recall@1/5/10)|T→I(Recall@1/5/10)|
> > > |------|--------------------------|--------------------------|
> > > |DECL|35.5/62.4/72.4|35.3/62.9/72.9|
> > > |RSH†|37.1/63.9/74.7|37.5/64.3/74.7|
> > >
> > > > **Q2: Computational Efficiency.**
> > >
> > > **R:**
> > > Table 2 summarizes the results from Table 3 of the main paper, with an additional single-feature triplet-loss baseline for reference. All training and encoding times are measured over 100 epochs.
> > >
> > > *Table 2. Comparison of the computational efficiency of RSH and baselines with 64 bits on MS-COCO under 20% noise.*
> > > |Method|FLOPs/Params/Train-time/Encode-time|
> > > |------|---------------------|
> > > |DSTH|5.579G/151.82M/1.31h/1.98s|
> > > |NRCH|5.578G/151.29M/0.52h/1.54s|
> > > |RSHNL|5.579G/151.59M/0.63h/2.74s|
> > > |DPBE|5.579G/152.53M/0.68h/2.83s|
> > > |DUaPH|5.578G/151.59M/2.51h/4.02s|
> > > |Triplet-loss|5.578G/151.24M/0.51h/1.74s|
> > > |RSH|5.578G/151.29M/0.51h/1.68s|
> > >
> > > Under a shared backbone, RSH and the baselines are nearly identical in FLOPs. Compared with a standard single-representation baseline, triplet-loss has 151.24M parameters, and RSH has 151.29M; the difference comes only from the uncertainty branch. Thus, the gains of RSH are not from a larger model, but from better similarity modeling under almost the same computational complexity.
> > >
> > > > **Q3: Alternative Ways of Fusing Semantic and Uncertainty Information.**
> > >
> > > **R:**
> > > We explored such alternatives in the ablation study. First, for pairwise uncertainty construction, RSH-U replaces $||u^I + u^T||_2$ with $||u^I||_2 + ||u^T||_2$, thus ignoring uncertainty correlation at the sample-pair level; it performs worse than RSH on all three datasets. Second, for the fusion form in SSM, RSH-O uses an alternative similarity metric that enlarges semantic separation for highly uncertain pairs instead of weakening semantic discrepancy; it also underperforms RSH. These results indicate that the current design, namely, adaptively weakening semantic discrepancy through relative uncertainty, is not arbitrary but more effective than other plausible fusion schemes.
> > >
> > > In the previous rebuttal, we also compared a two-layer MLP and a residual uncertainty branch as the uncertainty encoder. More complex branches can achieve comparable or slightly higher MAP in a few cases, but also introduce more parameters and longer training time. Hence, the current FC-based design provides a more reasonable performance-efficiency trade-off. The key contribution is therefore not the FC layer itself, but how uncertainty is incorporated into the similarity function and used to regulate optimization.
> > >
> > > > **Q4: Limitations and Potential Impact.**
> > >
> > > **R:**
> > > We accept this suggestion and will make it explicit in the final version. We will add a dedicated limitations subsection to state that the paper mainly evaluates controlled noisy-label settings, and that parameter behavior in large-scale scenarios still deserves further study. We will also briefly discuss potential societal risks.
> > >
> > > In summary, RSH provides a general mechanism that directly incorporates uncertainty into similarity learning and suppresses the influence of noisy pairs at the optimization level. The additional results on asymmetric noise, real-world noise, alternative fusion schemes, more complex encoders, and corrected efficiency comparisons are all intended to validate this point. We would be very grateful if you could reconsider your assessment of the overall contribution and maturity of the paper.

---

### Official Review · Reviewer_S7cq · 2026-03-19

**Soundness:** 3
**Presentation:** 3
**Significance:** 4
**Originality:** 4
**Overall Recommendation:** 6
**Confidence:** 5

**Summary:**

This paper addresses the widespread issue of noisy labels in cross-modal retrieval by proposing a Robust Self-reflective Hashing (RSH) framework. The core idea is to introduce uncertainty modeling on top of traditional semantic representations and design a Self-reflective Similarity Metric (SSM). By adaptively softening semantic discrepancies based on uncertainty level, the method mitigates the impact of noise or semantic ambiguity during model training. Unlike existing uncertainty modeling methods, which still rely on traditional similarity metrics for optimisation, resulting in an inability to effectively suppress noise sample, this paper directly incorporates uncertainty into the similarity calculation process. This enables adaptive adjustment at the optimisation level. This approach is highly novel and insightful. Additionally, the method exhibits good generalizability and can be integrated into various baselines. Experiments conducted on multiple datasets and under different noise ratios validate the effectiveness and robustness of the method.

**Compliance With Llm Reviewing Policy:**

Affirmed.

**Final Justification:**

I have read the rebuttal and other reviews. I believe the proposed method highly novel and innovative. Its key idea of incorporating uncertainty into similarity computation to suppress noise interference at the optimization level is both original and well motivated. I would therefore strongly recommend this paper for acceptance.

**Key Questions For Authors:**

- Is there a more rigorous theoretical explanation for the approximation from the discrete (indicator) form to the exponential continuous form in reflexive similarity?
- How stable are the parametric reflexive bias term and the semantic difference smoothing coefficient across different datasets?
- Are uncertainty features completely excluded during the inference stage? If so, would they introduce additional overhead or improve performance?

**Limitations:**

yes

**Strengths And Weaknesses:**

**1. Soundness:**

Strengths:
 - The Self-Reflective Similarity Metric (SSM) has a clear mathematical formulation, and its suppression effect on samples with high uncertainty is explained through gradient analysis, making it theoretically sound.
- The method forms a complete workflow, ranging from uncertainty modeling to similarity calculation and optimization, and the overall design exhibits good consistency.

Weaknesses:
- The derivation from the indicator function to the continuous approximation (Eq. 7–Eq. 9) is brief and could be further elaborated theoretically.
- The explanations for certain formulas (such as the dual feature representation) could be expanded to enhance rigour.
- Some symbols and variable definitions are rather concise. Additional explanations should be provided to help people understand them.

**2. Presentation:**

Strengths:
- The overall structure is clear, with distinct sections for problem motivation, method design and experimental validation. Illustrations (e.g. figures 1–4) aid understanding of the core ideas.

Weaknesses:

- Some paragraphs contain redundant descriptions (e.g. the discussion of noisy labels in the introduction), which could be streamlined to enhance coherence.
- The boundaries of responsibility between different modules (such as DFR and SSM) should be further clarified to enhance the method's comprehensibility.

**3. Significance:**

Strengths:
- It addresses the practical problem of noisy labels, offering strong practical value, particularly in large-scale weakly labelled and noisy label scenarios.
- The method is highly plug-and-play and can easily be integrated into existing cross-modal hashing frameworks.

Weaknesses:

The method's improvements are primarily in terms of robustness, and its advantages in extremely low-noise scenarios are relatively limited.

**4. Originality:**

Strengths:
- The approach of directly incorporating uncertainty into the similarity metric, rather than solely using it for weighting or modelling distributions, is novel.
- It is clearly distinct from existing uncertainty modelling and noise-resistant methods.

---

> ### Author Rebuttal · Authors · 2026-03-31
>
> We sincerely thank the reviewer for the constructive comments and helpful suggestions.
>
> > **Q1: Theoretical Explanation of the Transition from Indicator Functions to Continuous Approximations.**
>
> **R1:** Equations (7)-(9) transform the discrete indicator function into the continuous form $\exp(-d/\tau)$ to approximate the $0$ - $1$ semantic constraint, where $d$ represents the semantic difference. This function maintains monotonicity: as $d \to 0$, it approaches 1, indicating semantic consistency; as $d$ increases, it decays continuously, achieving soft constraints. Building on this, Equations (8)–(9) introduce uncertainty by reparameterizing the semantic difference as $\tilde{d}$, yielding $\exp(-\tilde{d}/\tau)$, thereby simultaneously encoding both semantic difference and uncertainty. This continuous formulation ensures differentiable optimization while preserving the monotonic consistency of the original decision boundary, and modulates the contributions of similarity and gradients through uncertainty.
>
> > **Q2: Parameter Stability.**
>
> **R2:** The stability of the self-reflective bias term and the semantic softening coefficient has been systematically verified under different noise ratios. As shown in Table 1, these parameters remain stable across a relatively wide range of values and do not require extensive fine-tuning. This property stems from the fact that the parameters mainly regulate the sensitivity of similarity responses rather than directly determining the decision boundary. In addition, the uncertainty-aware SSM dynamically adjusts the semantic discrepancy strength according to the data distribution, thereby reducing reliance on fixed hyperparameters.
>
> *Table 1.Parameter analysis of average MAP results on MIRFLICKR-25K and IAPR TC-12 with 128 bits under different noise.*
> |$\gamma/\tau$|0/1|1/3|2/5|3/7|4/9|5/11|6/13|
> |---|---|---|---|---|---|----|----|
> |50% noise, MIRFLICKR-25K|0.8436/0.8437|0.8429/0.8449|0.8457/0.8484|0.8464/0.8465|0.8456/0.8464|0.8435/0.8499|0.8433/0.8497|
> |80% noise, MIRFLICKR-25K|0.8281/0.7833|0.8212/0.8210|0.8226/0.8212|0.8254/0.8223|0.8208/0.8254|0.8219/0.8240|0.8230/0.8213|
> |50% noise, IAPR TC-12|0.7021/0.7010|0.7040/0.7027|0.7045/0.7037|0.7048/0.7046|0.7048/0.7048|0.7032/0.7035|0.7034/0.7026|
> |80% noise, IAPR TC-12|0.7010/0.7060|0.7021/0.7047|0.7041/0.7057|0.7063/0.7061|0.7056/0.7063|0.7042/0.7049|0.7029/0.7030|
>
>
> > **Q3: Use of Uncertainty Features in the Inference Stage.**
>
> **R3:** During the inference stage, retrieval relies solely on the learned semantic hash representations and does not depend on the uncertainty branch; therefore, it does not introduce additional storage or computational overhead. Uncertainty modeling is involved only in the training stage for similarity learning, where it modulates the gradient contributions of sample pairs to suppress noise interference and enhance representation robustness. As shown in Table 2, incorporating it into the inference stage does not yield improvement; the current design strikes a more reasonable balance between performance and efficiency.
>
> *Table 2. Performance comparison of average MAP scores on MIRFLICKR-25K under 50% noise, w/ or w/o incorporating uncertainty features during inference.*
> |Method|32 bits|64 bits|128 bits|
> |------|--|--|---|
> |w/ Inference|0.7961|0.8155|0.8179|
> |w/o Inference|0.8177|0.8338|0.8464|
>
> > **Q4: Limited performance improvement.**
>
> **R4:**
> RSH is intended to improve robustness in noisy-label scenarios rather than to maximize performance under ideal clean-data conditions. As shown in Table 1 of the paper, its advantage becomes increasingly evident as the noise ratio rises, suggesting that it can effectively mitigate the adverse influence of noisy labels. In addition, as a plug-and-play similarity metric mechanism, RSH incurs only negligible overhead and still brings consistent improvements over strong baselines when used with the original base loss. Figure 5 of the paper further shows that replacing the original similarity metric with RSH consistently improves the robustness of different cross-modal hashing methods. Statistical significance results under low-noise settings are provided in our response to [Reviewer UVNh](https://openreview.net/forum?id=jyGcO0W5xd¬eId=1cbxjr2QZb), Q1.

---

> > ### Author Rebuttal · Reviewer_S7cq · 2026-04-01
> >
> > I have read the rebuttal and other reviews. I appreciate the rebuttal. My concerns are resolved by the rebuttal. In particular, I believe the proposed method highly novel and innovative. Its key idea of incorporating uncertainty into similarity computation to suppress noise interference at the optimization level is both original and well motivated. I would therefore strongly recommend this paper for acceptance.

---

> > > ### Author Response · Authors · 2026-04-06
> > >
> > > We sincerely thank the reviewer for the encouraging feedback and for taking the time to carefully read both the rebuttal and the other reviews. We greatly appreciate your recognition of the novelty and motivation of our method. Thank you again for your time and consideration.

---

### Decision · Program_Chairs · 2026-04-30

**Decision:**

Accept (regular)

**Comment:**

This paper addresses cross-modal retrieval under noisy supervision and proposes a self-reflective similarity mechanism that incorporates uncertainty directly into similarity computation. The method is lightweight, conceptually clear, and designed to be easily integrated into existing hashing frameworks. Experimental results demonstrate improved robustness under noisy-label settings.

The reviewers agree that the problem is relevant and the method is technically sound, but differ in their assessment of novelty strength and empirical scope. Positive evaluations highlight the principled formulation and consistent robustness improvements, while more critical assessments focus on the magnitude of gains, evaluation breadth, and theoretical generalization.

The rebuttal provided additional clarifications and experimental evidence that addressed several technical concerns. At least one reviewer indicated that the main issues were resolved and signaled an increased score, while some reservations regarding impact and generality remain.

Overall, the paper appears technically correct and provides a meaningful, though incremental, contribution to robust cross-modal retrieval. The remaining concerns relate primarily to scope and maturity rather than correctness.

Recommendation: Weak Accept.
The submission meets the conference standard for publication, and the rebuttal improved confidence in the validity of the method. The authors are encouraged to incorporate the clarifications and limitations discussed during the review process into the final version.